# A High MCM6 Proliferative Index in Atypical Meningioma Is Associated with Shorter Progression Free and Overall Survivals

**DOI:** 10.3390/cancers15020535

**Published:** 2023-01-16

**Authors:** Guillaume Gauchotte, Charles Bédel, Emilie Lardenois, Sébastien Hergalant, Laura Cuglietta, Robin Pflaum, Stéphanie Lacomme, Héloïse Pina, Mathilde Treffel, Fabien Rech, Shyue-Fang Battaglia-Hsu

**Affiliations:** 1Department of Biopathology, CHRU Nancy, Université de Lorraine, F-54500 Vandoeuvre-lès-Nancy, France; 2Centre de Ressources Biologiques, BB-0033-00035, CHRU Nancy, F-54500 Vandoeuvre-lès-Nancy, France; 3INSERM U1256, NGERE, F-54500 Vandoeuvre-lès-Nancy, France; 4Faculty of Medicine, Université de Lorraine, F-54500 Vandoeuvre-lès-Nancy, France; 5CNRS, CRAN, Université de Lorraine, F-54000 Nancy, France; 6Department of Neurosurgery, CHRU Nancy, F-54035 Nancy, France

**Keywords:** MCM6, Ki-67, PTEN, proliferation, atypical meningioma, meningioma

## Abstract

**Simple Summary:**

The aim of this study was to evaluate the prognostic value of MCM6 relative to that of Ki-67 in a series of grade 1 (World Health Organization 2021; *n* = 100) and grade 2 (atypical) meningiomas (*n* = 69), using immunohistochemistry, and to evaluate its correlation with methylation classes. In a multivariate model, the LI (Labeling Index) of MCM6 correlated with progression free survival of grade 2, but not grade 1 meningiomas. MCM6 was also correlated with overall survival in univariate analysis. No correlation was found with the methylation classes and subclasses returned by the meningioma algorithm MNGv2.4. We found a significant correlation between PTEN loss and high MCM6 or Ki-67 LI. Our evidence here suggests that MCM6 is a relevant and reproducible marker in atypical meningiomas. It is also easy-to use and could also allow to identify a highly aggressive subtype of proliferative meningiomas.

**Abstract:**

The aim of this study was to evaluate the prognostic value of MCM6, in comparison with Ki-67, in two series of grade 1 and 2 meningiomas, and to evaluate its correlation with methylation classes. The first cohort included 100 benign (grade 1, World Health Organization 2021) meningiomas, and the second 69 atypical meningiomas (grade 2). Immunohistochemical Ki-67 and MCM6 labeling indices (LI) were evaluated independently by two observers. Among the atypical meningiomas, 33 cases were also studied by genome-wide DNA methylation. In grade 2 meningiomas, but not grade 1, both Ki-67 and MCM6 LIs were correlated with PFS (*p* = 0.004 and *p* = 0.005, respectively; Cox univariate analyses). Additionally, MCM6 was correlated with overall survival only in univariate analysis. In a multivariate model, including mitotic index, Ki-67, MCM6, age, sex, and the quality of surgical resection, only MCM6 was correlated with PFS (*p* = 0.046). Additionally, we found a significant correlation between PTEN loss and high MCM6 or Ki-67 LIs. Although no correlation was found with the methylation classes and subtypes returned by the meningioma algorithm MNGv2.4., MCM6 LI was significantly correlated with the methylation of 2 MCM6 gene body loci. In conclusion, MCM6 is a relevant prognostic marker in atypical meningiomas. This reproducible and easy-to-use marker allows the identification of a highly aggressive subtype of proliferative meningiomas, characterized notably by frequent PTEN losses, which was previously reported to be sensitive to histone deacetylase inhibitors.

## 1. Introduction

Meningioma is one of the most common central nervous system tumors, accounting for one third of all reported brain tumors [1]. Most meningiomas are benign and require minimum intervention if they are restricted and with no or minor symptom, but it is imperative to identify the infrequent recurrent low grade meningiomas bearing high risk of progression. Higher grade meningiomas grow rapidly and can invade adjacent tissues. The histologic grading system, established by World Health Organization (WHO) in 2021 based on mitotic rate, cytoarchitectural features, brain invasion, histologic subtype, and/or molecular alterations (TERT, CDKN2A/B), is strongly correlated with progression-free and overall survival [2]. However, the WHO grade is not sufficient to accurately predict the risk of progression. This is particularly challenging for patient care as meningioma treatment options are generally based on the WHO grades: while asymptomatic grade 1 requires no immediate treatment, the therapeutic guideline for both atypical WHO grade 2 and anaplastic WHO grade 3 is surgery removal often followed by radiotherapy [3].

To resolve the difficulty in the identification of meningiomas with high risk of recurrence, we previously examined the prognostic values of several cell cycle related markers including MCM6, Ki-67, PHH3, cyclin D1, and p53 in cohorts of grade 1 (*n* = 32) and grade 2 meningiomas (*n* = 27) [4]. We identified both Ki-67 and MCM6 as potentially adequate prognostic markers. Both markers showed positive correlation with WHO grading and progression free survival (PFS), with high inter-observer reproducibility. The usefulness of Ki-67 has indeed been widely reported in numerous tumors as prognostic marker, including meningiomas [5,6]. However, the nuclei labeling of this marker may be unreliable, as inter-laboratory differences in both intensity and frequency were reported [7]. These variations might partly be due to the different methods used during the processes of antigen retrieval and staining [8]. We suggested that these variations may also be a consequence of its low labeling index. The MCM6 protein is a part of the DNA helicase replicative complex, composed of six distinct but related polypeptides, MCM2, MCM3, MCM4, MCM5, MCM6, and MCM7. It functions by unwinding the two DNA spiral strands [9] and by promoting G1/S transition [10]. Many studies have shown that the level of expression of MCM6, basing on mRNA detection or immunohistochemistry, was correlated with clinical outcome, e.g., in gliomas, renal cell carcinoma, endometrial adenocarcinoma, lung cancer, osteosarcoma, pheochromocytoma, neuroblastoma, and meningioma [4,10,11,12,13,14,15,16,17,18,19,20]. Our previous evidence singled out MCM6 as an efficient marker by virtue of its dominant presence in proliferating cells that resulted in striking clear-cut differences in MCM6 labeling index (LI) of the indolent vs. the recurrent meningiomas, with a high reproducibility [4]. However, in this preliminary previous study, the performances of these markers within grade 1 or grade 2 meningiomas were not evaluated.

In two distinct cohorts of grade 1 and grade 2 meningiomas, we aimed to better clarify the prognostic role of Ki-67 and MCM6, and thus their potential usefulness in clinical practice within grade 1 or grade 2 tumors, in addition to the WHO grading system. Additionally, we intended to correlate these markers with methylation classes and main copy number alterations based on methylome assay.

## 2. Materials and Methods

### 2.1. Population, Clinical Data, and WHO Grading

One hundred sixty-nine cases of intracranial meningiomas were retrospectively retrieved from the Biobank of the Department of Pathology at CHU-Nancy, France. These included a cohort of 100 consecutive benign meningiomas (grade 1) operated between 2002 and 2003, and 69 consecutive atypical meningiomas (grade 2) from 2001 to 2018. Patient anonymity was strictly respected, following local ethical guidelines. Clinical data for individual patients, including age, sex, time to progression or death, the quality of resection, and treatment, were retrospectively collected from the Department of Neurosurgery of the same hospital.

Diagnosis and histologic grade, according to the criteria of the 2021 WHO classification, were reviewed by 2 experienced pathologists. Specifically for grade 2, these included brain invasion, and/or 4 or more, but fewer than 20 mitoses per 10 high-power fields (1.6 mm^2^), and/or 3 or more of the following: increased cellularity, small cell change, prominent nucleoli, loss of lobular architecture (sheeting), necrosis (without preoperative embolization). In case of grading disagreement, a consensus was reached after collegial discussion.

### 2.2. Immunohistochemistry

A representative paraffin block from each case was selected for this study. When several were available, the block with the higher cell density areas was chosen. Each paraffin section analyzed came from the most significant block judged by the density of the MCM6 and Ki-67 markings and the quality of the sampling. Paraffin sections were prepared as follows: sections of 5 µm thickness were first placed on a glass slide and then immersed in the Dako PT LINK automat with a sodium citrate buffer (pH 6) for dewaxing and antigen unmasking. Two primary antibodies were used: Ki-67 (1/200; mouse monoclonal, MIB-1, Dako Cytomation, Glostrup, Denmark) and MCM6 (1/400; goat poly- clonal, Santa Cruz Biotechnology, Heidelberg, Germany). The antigen–antibody reaction was performed with Dako Autostainer Plus (Dako) using biotin-streptavidin amplification and diaminobenzidine as a chromogen. Additionally, to evaluate the inter-laboratory reproducibility of Ki-67 and MCM6, a subset of 47 randomly selected cases were analyzed in another pathology laboratory (Biopathology, Institut de Cancérologie de Lorraine), using the same primary antibodies, on a Roche Ventana Benchmark Ultra automate.

The count was performed under an optical microscope under a magnification of 400×, and the percentage of markers assigned to each tumor was made on a field of 1000 cells in the densest staining area of the slide, by two observers. A mean value was then calculated. The labeling index (LI) of each tumor was defined as the percentage of cells displaying nuclear expression (any clear-cut nuclear labeling was considered as sufficient, independently of intensity).

### 2.3. Methylome

Frozen meningioma tissue samples were available in 33 cases of atypical meningiomas. DNA was extracted with Macherey Nagel DNA extraction kit (Macherey-466 Nagel, Düren, Germany). After qualitative control, 900 ng of the extracted DNA were used to perform analyses using Illumina MethylationEPIC BeadChip following the manufacturer instructions. Raw data files (IDAT) were generated and used for downstream bioinformatics under the minfi package, with R v4.1. Normalization was carried out following the FunNorm procedure [21].

The IDAT files were then submitted to the brain tumor classification, copy-number variation (CNV) estimation, and the meningioma algorithm, according to Capper et al. [22] and Sahm et al. [23]. Raw methylation files (IDAT) were uploaded on the MolecularNeuropathology.org server using the v11b4 classifier for brain tumor classification and MNGv2.4 for meningioma subtype classification. The evaluation of copy-number aberrations (along with CNV plots), CDKN2A/B loss and PTEN loss were extracted from the CNV profile section of the generated report. Spearman’s correlation tests were used for correlation analyses between *MKI67* and *MCM6* CpG methylation levels and Ki-67 and MCM6 LI, adjusted for false discovery rate (FDR) following the Benjamini–Hochberg procedure, as previously reported [24].

### 2.4. Statistical Analysis

Statistical analyses were performed using MedCalc and IBM SPSS. Non-parametric tests were used to evaluate the correlations between quantitative and/or qualitative variables, including Spearman, Mann–Whitney–Wilcoxon, or Kruskall–Wallis tests. For the survival data, we first looked for an optimal threshold with the Cutoff Finder tool [25]. Progression-free survival (PFS) and, in grade 2, overall survival (OS) analyses were performed with the Kaplan-Meier estimation (Log-rank test). Cox models were also used for PFS, using LI as quantitative variables. The cubic spline method was used to detect eventual time-dependent and non-linear effects. The level of agreement was measured with the intraclass correlation coefficient (ICC) for quantitative variables. A *p*-value ≤ 0.05 was considered statistically significant.

## 3. Results

### 3.1. Grade 1 Meningiomas

The grade 1 meningioma cohort included 100 patients aged 29–79 y.o. (mean age: 54 y.o.), with a male to female ratio of 0.2, and involved in most cases skull base (48/100) or convexity (32/100) (Table 1). A progression occurred within five years in 15% of cases. No significant correlation was found between PFS and age, sex, or the extent of surgery. The median LI was 1.4% for Ki-67 and 16% for MCM6. The Spearman test revealed that these 2 markers were significantly correlated (rho = 0.348: *p* < 0.0001). The log-rank test indicated that a Ki-67 LI greater than 1.4% was significantly correlated with a shorter PFS (*p* = 0.02) (Figure 1a). The same test showed an absence of correlation between MCM6 and PFS (*p* > 0.5) (Figure 1b). No significant correlation was found with the Cox Model using Ki-67 and MCM6 as quantitative variables (*p* > 0.5).

### 3.2. Grade 2 Meningiomas

In the grade 2 (atypical) meningioma cohort (*n* = 69), mean age was 62 y.o., with a male to female ratio of 0.86. All the cases involved were cranial meningiomas, including in most cases convexity (46/69) or skull base (12/69). In addition, 56.5% of these were irradiated (adjuvant radiotherapy). Mean PFS was 22.3 months after surgery.

In atypical meningiomas, the median Ki-67 (Figure 2a,c) and MCM6 (Figure 2b,d) LI were 20% and 61%, respectively. The Spearman test showed that Ki-67 and MCM6 were significantly correlated (rho = 0.375: *p* = 0.0015). Furthermore, Ki-67 and MCM6 were both correlated with the mitotic index (rho = 0.284, *p* = 0.02; rho = 0.294, *p* = 0.01, respectively).

Log-Rank test on survival indicated that a grade 2 meningioma with a Ki-67 equal or greater than 30% was associated with a shorter PFS (*p* = 0.002, HR = 3.71) (Figure 1c). Likewise, Log-Rank test predicted a shorter PFS associated with patients having a grade 2 meningioma with a MCM6 LI equal or greater than 50% (*p* = 0.001, HR = 3.05) (Figure 1d). Additionally, survival analyses showed that a high MCM6 LI was also correlated with a shorter OS, at the 55% threshold (*p* = 0.02). No significant threshold was found for Ki-67 LI regarding OS.

We further cross-checked the correlations for PFS using Cox models, with Ki-67, MCM6, and mitotic index as quantitative variables. In univariate setup, positive correlation was found between Ki-67 and PFS (*p* = 0.004; HR = 1.04) as well as between MCM6 and PFS (*p* = 0.005; HR = 1.03). No correlation could be found between mitotic index and PFS (*p* = 0.37). In multivariate setup, including mitotic index, Ki-67, MCM6, age, sex, and the quality of surgery as covariates, we were able to identify only a statically significant association between MCM6 and PFS (*p* = 0.046; HR = 1.02) since the *p*-value between Ki-67 and PFS did not reach the threshold of 0.05 (*p* = 0.0502, HR = 1.03). We also found that no variable was correlated with OS.

### 3.3. Correlation with Methylome and CNV

The global brain tumor classifier (MolecularNeuropathology.org) reported the methylation class “meningioma” with a valid score (calibrated score > 0.9) for all samples. Methylome classes were determined with the Sahm et al. online algorithm [23], in 33 cases of WHO grade 2 meningiomas: 15% (5/33) were classified as benign, 36% (12/33) as intermediate, and 6% (2/33) as malignant. The methylation class was not interpretable (score < 0.9) in 42% (14/33). Neither the LI of MCM6 nor that of Ki-67 correlated with the methylation class. Moreover, no correlation between methylation classes and survival (PFS and OS) was found. When excluding cases with a score lesser than 0.5 (4/33), seven cases were classified as benign, 18 as intermediate, and four as malignant, without correlation with LI (MCM6 or Ki-67) and survival (PFS: *p* = 0.90; OS: *p* = 0.32).

For *MKI67* we found no significant correlation (Appendix A) between Ki67 LI% and DNA methylation of the CpGs linked to *MKI67* promoter regions or within gene body. However, three consecutive island/promoter CpGs (cg26235537, cg10026577, cg02306970) were near significance after correction for the FDR and could constitute a VMR (variably methylated region) of positive correlation (Spearman’s rho = 0.5). For *MCM6*, we found two significant positive correlations within *MCM6* gene body (cg10512742 with rho = 0.66 and cg15903064 with rho = 0.5) (Appendix A). These two loci could be linked to transcription robustness and stability mechanisms, and may relate to an increase of transcription [26].

When considering the copy number alteration analysis, a *PTEN* loss was found in 42% (14/33) of cases. No homozygous deletion of *CDKN2A/B* was found. The loss of *PTEN* appeared to be correlated with both Ki-67 (*p* = 0.019) and MCM6 (*p* = 0.002) LI (Figure 3). No correlation with 22q/*NF2* loss was found.

### 3.4. Reproducibility and Evaluation of the Potential Influence of Time-Related Tissue Degradation

The inter-observer reproducibility (Table 2) was moderate for the mitotic index (ICC = 0.73) and good for Ki-67 and MCM6 (ICC = 0.85 and ICC = 0.87, respectively). Additionally, the evaluation of the inter-laboratory reproducibility showed a good reproducibility for Ki-67 as well as MCM6 (ICC = 0.81 and ICC = 0.87, respectively). Finally, in order to evaluate the potential influence of time-related tissue degradation, we compared the Ki-67 and MCM6 LI between different periods, in the grade 1 (2002 vs. 2003) and the grade 2 (before 2010 vs. 2010 and after) cohorts. No significant differences were found (*p* > 0.1).

## 4. Discussion

Before the present study, the specific prognostic value of MCM6 in atypical meningiomas was still not elucidated. Using 2 cohorts of 100 grade 1 and 69 grade 2 meningiomas, we compared the prognostic values of Ki-67 and MCM6. Our results showed that Ki-67, despite inter-laboratory variations often cited in the literature and a rather low LI in low grade tumors, remains an efficient prognostic marker for both grade 1 and grade 2 meningiomas. The particular value of Ki-67 LI in grade 1 meningioma as a prognosis marker of PFS should be underlined given the fact that at a threshold as low as 1.4%, Ki-67 was still correlated with shorter PFS (Log-Rank test). Previously, we have suggested that MCM6 may be a more suitable marker based on its higher expression [4]. However, in the present study, we evidenced that it is not applicable for grade 1 meningioma. For grade 2 meningioma, Ki-67 and MCM6 both appear adequate (≥30% for Ki-67, and ≥50% for MCM6, Log-Rank test) for the identification of tissues with potential shorter PFS. The multivariate Cox model revealed sole MCM6 to be correlated with PFS. For Ki-67, the statistical correlation was very close to the threshold, which could be merely due to the limited sample size. Additionally, MCM6 LI, but not Ki-67, was correlated with shorter OS (Log-Rank test). As the immunohistochemical technique used here is an economical, quick, and easy-to-use method, in comparison with other methods, such as mRNA quantification or DNA methylome, we suggest that Ki-67, MCM6 immunohistochemical staining can be applicable by the pathologist in daily practice, at the initial diagnosis, on formalin-fixed tissue blocks to efficiently and rapidly identify the high risk meningiomas. However, further multicentric studies are needed to corroborate these findings.

In this study, we also inspected the methylation level of the target genes, *MKI67* and *MCM6*, and the correlation with immunohistochemical staining (Ki-67 and MCM6 LI). No significant correlation was found for *MKI67*, but 3 consecutive island/promoter CpGs were near significance after correction for the FDR and could constitute a VMR of positive correlation. To our knowledge, a DMR (differentially methylated region) or VMR with positive correlation within a regulatory region could be linked to a tissue-specific mechanism and alternative transcription [27]. Interestingly, for *MCM6*, we found two significant positive correlations within MCM6 gene body. These two loci could be linked to transcription robustness and stability mechanisms, and thus to an enhanced transcription. This is an interesting finding which may explain in part the overexpression of MCM6. Furthermore, gene body methylation sites could constitute an unexpected therapeutic target for DNA methylation inhibitors [26].

Recently, in a multi-omics study, Nassiri et al. identified a highly aggressive subgroup of meningioma, referred as the MG4 proliferative group, characterized by a high fraction of disrupted genome, frequent PTEN loss, high MCM expression, poor clinical outcome, and sensitivity to vorinostat, a histone deacetylase inhibitor [28]. Our results are in line with this study, showing an association of high MCM6 LI with PTEN loss and poorer survival. Hence, MCM6 could also be a predictive marker of response to histone deacetylase inhibitors. Nassiri et al. hypothesized that vorinostat would be able to target several critical pathways that were specifically upregulated in these proliferative meningiomas. Indeed, vorinostat selectively decreased the viability of cell lines derived from patients with MG4 tumors, and not cell lines derived from patients with tumors belonging to other molecular groups. Similar results were found in mice with intracranial xeno-grafts of patient-derived MG4 cell lines. The authors reported that vorinostat was able to target MYC, E2F, G1 transition, mitotic spindle, nucleosome, nuclear division, and mitotic protein regulation [28]. Interestingly, previous data reported that histone acetylation promotes MCM loading via enhanced chromatin accessibility [29] and that MCM-2 may be a target of trichostatin, a classical histone deacetylase (HDAC) inhibitor [30]. In addition to its usefulness as a prognostic and predictive marker, MCM6 could also be a therapeutical target, as suggested by few experimental studies. In neuroblastoma, in a pre-clinical cell model, proliferation, migration, and invasion were significantly inhibited after MCM6 was interfered by siRNA [10]. In hepatocellular carcinoma, the knockdown of MCM6 significantly decreased proliferative and migratory and invasive capability of cancer cells in vitro, as well as decreased tumor volume, weight and the number of pulmonary metastases in vivo [31]. In neuroendocrine prostate cancer (NEPC), MCM2-7 could be inhibited by a treatment with ciprofloxacin: the authors showed that this treatment was able to inhibit NEPC cell proliferation and migration in vitro, significantly delaying NEPC tumor xenograft growth, and to partially reverse the neuroendocrine phenotype in vivo [32]. Recently, in gastric cancer, Wang et al. identified purpureaside C as a novel MCM6 inhibitor, and showed that, in vitro, this treatment suppressed gastric carcinoma cell growth and synergized with 5-fluorouracil to induce cell death [33]. Further pre-clinical studies are needed to confirm these promising results, notably in high grade meningiomas.

## 5. Conclusions

In conclusion, MCM6 immunohistochemical LI is a relevant prognostic marker in atypical meningiomas, significantly correlated with PFS and OS. This reproducible and easy-to use marker could also allow to identify a highly aggressive subtype of proliferative meningiomas, characterized by frequent PTEN losses, which was previously reported to be sensitive to histone deacetylase inhibitors. It may also allow to select patients for specific therapies targeting the MCM complex.

## Figures and Tables

**Figure 1 cancers-15-00535-f001:**
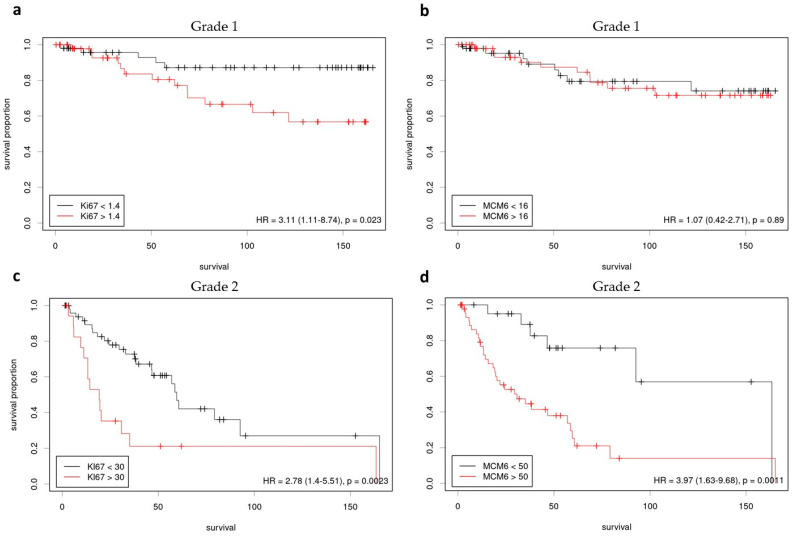
Survival analyses, progression free survival (Log-Rank, Kaplan-Meier curves; thresholds determined with Cutoff finder [25]): (**a**) Ki-67 in grade 1 meningiomas; (**b**) MCM6 in grade 1 meningiomas; (**c**) Ki-67 in grade 2 meningiomas; (**d**) MCM6 in grade 3 meningiomas.

**Figure 2 cancers-15-00535-f002:**
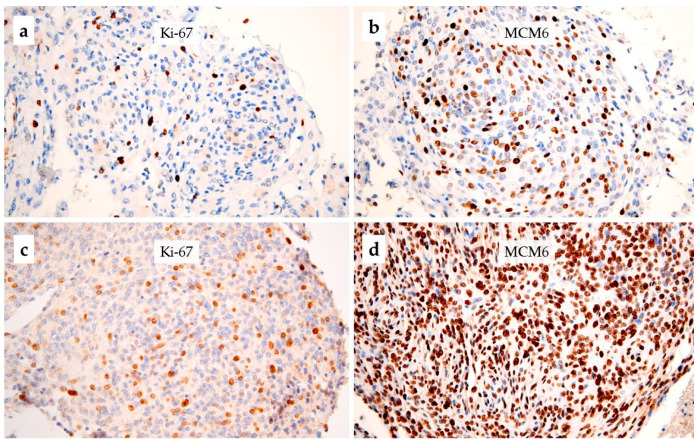
Immunohistochemistry for Ki-67 and MCM6 in two cases of grade 2 meningioma: (**a**,**b**) moderate staining for Ki-67 and MCM6 in a non-relapsing meningioma, respectively; (**c**,**d**) high staining for Ki-67 and MCM6, respectively, in a case of atypical meningioma, that recurred 4 months after the initial excision (×200 magnification).

**Figure 3 cancers-15-00535-f003:**
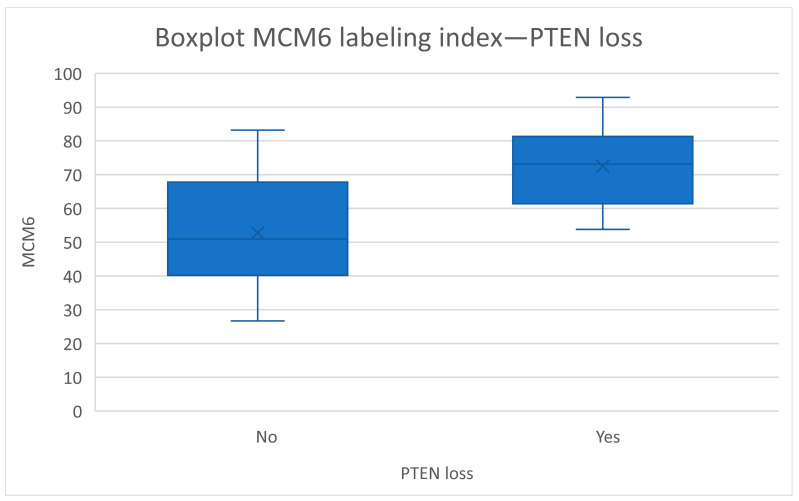
Correlation between PTEN loss and MCM6 labeling index (*p* = 0.002, Mann-Whitney Wilcoxon).

**Table 1 cancers-15-00535-t001:** Clinical data.

Variable	Grade 1 *n* = 100	Grade 2 *n* = 69
Age (mean; min.-max.)	54 (29–79) y.o.	62 (33–91) y.o.
Sex	F: 83 M: 17	F: 39 M: 30
Gross total resection	86%	62%
Localization	Convexity: 32 Falcorial/parafalcorial: 15 Skullbase: 48 Spinal: 5	Convexity: 46 Falcorial/parafalcorial: 8 Skullbase: 12 Ventricular: 3
Adjuvant radiotherapy	8%	56%
Progression	18%	51%
Death	11%	21%

**Table 2 cancers-15-00535-t002:** Inter-observer and inter-laboratory reproducibility (intraclass correlation coefficient).

Marker	Inter-Observer	Inter-Laboratory
Ki-67	0.85	0.81
MCM6	0.87	0.87

## Data Availability

The data can be shared up on request.

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
