# Peer review of "A High MCM6 Proliferative Index in Atypical Meningioma Is Associated with Shorter Progression Free and Overall Survivals"

_cancers, 2023, doi:10.3390/cancers15020535_

Round 1
Reviewer 1 Report
A very interesting and relevant report.
Grammar/English/punctuation etc excellent with no errors / corrections noted.
In Discussion, please address in more detail how MCM6 may indicate responsiveness to HDAC inhibitor vorinostat - this needs to be more clearly outlined.
Author Response
A very interesting and relevant report.
Grammar/English/punctuation etc excellent with no errors / corrections noted.
In Discussion, please address in more detail how MCM6 may indicate responsiveness to HDAC inhibitor vorinostat - this needs to be more clearly outlined.
Thank very much for your comments. We added in the discussion more data about vorinostat and the potential links between HDAC and MCM complex (added pages 8-9, lines 319-328).
Reviewer 2 Report
Gauchotte et al. investigated the prognostic role of MCM6 in a retrospective cohort of grade 1 and 2 meningiomas from their institution. The study stems from a previous work on the same topic suggesting a potential interest of this marker. The article is clear and well-written. I have just a few concerns and suggestions:
- I would suggest to move to the introduction the first part of the discussion, which actually sets the background for the study
- Grade 1 meningiomas samples were collected between 2002 and 2003. Can the authors comment on the potential influence of time-related factors (e.g., tissue degradation) on the absence of prognostic significance observed for MCM6 in this patient subset?
- can the authors specify how the tresholds for Ki67 and MCM6 stratification (see Fig. 1) was obtained? do these threshold correspond to median values?
- It seems that MCM6 was correlated with OS only in univariate and not in multivariate analyses; this should be clearly stated in the abstract and in the discussion/conclusion to vaoid any confusion. In addition, the correlation between Ki67 values and PFS was not statistically significant but was very close to the threshold, and this could be merely due to the limited sample size. In the light of these results, I would suggest much more caution when emphasizing the added value of MCM6 compared to Ki67 since the data at hand are quite slim. Further multicentric studies are needed to corroborate these findings.
Author Response
Gauchotte et al. investigated the prognostic role of MCM6 in a retrospective cohort of grade 1 and 2 meningiomas from their institution. The study stems from a previous work on the same topic suggesting a potential interest of this marker. The article is clear and well-written. I have just a few concerns and suggestions:
- I would suggest to move to the introduction the first part of the discussion, which actually sets the background for the study
Thank you for your comments. The first part of the discussion, about MCM6, was moved to the introduction (page 3, lines 70-77).
- Grade 1 meningiomas samples were collected between 2002 and 2003. Can the authors comment on the potential influence of time-related factors (e.g., tissue degradation) on the absence of prognostic significance observed for MCM6 in this patient subset?
We did not observe any significant degradation of the signal in oldest samples. In order to evaluate the potential influence of time-related tissue degradation, we compared the Ki-67 and MCM6 LI between different periods. In grade 2 samples, we did not find any significant difference between samples collected before and after 2010, both for MCM6 and Ki-67 (P > 0.10). Similar results were found in grade 1, 2002 vs. 2003 (P > 0.10) (added in results, page 7, lines 258-262).
- can the authors specify how the tresholds for Ki67 and MCM6 stratification (see Fig. 1) was obtained? do these threshold correspond to median values?
The thresholds were determined with the Cutoff finder tool (page 4, line 154) (added in Figure 1 legend, page 5, line 182).
- It seems that MCM6 was correlated with OS only in univariate and not in multivariate analyses; this should be clearly stated in the abstract and in the discussion/conclusion to vaoid any confusion. In addition, the correlation between Ki67 values and PFS was not statistically significant but was very close to the threshold, and this could be merely due to the limited sample size. In the light of these results, I would suggest much more caution when emphasizing the added value of MCM6 compared to Ki67 since the data at hand are quite slim. Further multicentric studies are needed to corroborate these findings.
We added that correlation was found only in univariate analyses both in the simple summary and abstract, page 1 (lines 20 and 32-33).
Added in the discussion section: “The statistical correlation was for Ki-67 very close to the threshold, which could be merely due to the limited sample size” (page 8, lines 291-292); “Further multicentric studies are needed to corroborate these findings” (page 8, lines 298-299).
Reviewer 3 Report
The study focuses on the inspection of KI67 and MCM6 genes in the meningioma grade1 and grade2 tumors by measurement of their labelling index (LI) as measurement in proportion of cells with these genes highly active. Clear correlation between both genes’ LI values was identified as well as negative association with survival (MCM6 for grade1, Ki67 for both grades). Interestingly, no correlation was found to methylation derived classification.
Specific reviewer comments:
- The methylation classification is driven by large amount of highly variable CpG cites across genome, however there could be also direct inspection of methylation level in target genes. Are there any CpG cites in promoter and gene body of MCM6 and KI67 that demonstrate correlation to LI values for these genes?
- The methylation class was not interpretable for quite large amount of samples (42%). Did you try to inspect lower filtering cut limit, e.g. 0.5? Also, how about global classifier MNP v11/12, were all tumor samples distinguished as meningioma?
- Figures 1&2 subparts lack suffixes in the text, e.g. Figure 1a,b,...
Author Response
The study focuses on the inspection of KI67 and MCM6 genes in the meningioma grade1 and grade2 tumors by measurement of their labelling index (LI) as measurement in proportion of cells with these genes highly active. Clear correlation between both genes’ LI values was identified as well as negative association with survival (MCM6 for grade1, Ki67 for both grades). Interestingly, no correlation was found to methylation derived classification.
Specific reviewer comments:
- The methylation classification is driven by large amount of highly variable CpG cites across genome, however there could be also direct inspection of methylation level in target genes. Are there any CpG cites in promoter and gene body of MCM6 and KI67 that demonstrate correlation to LI values for these genes?
Thank you for this very relevant comment which lead to new interesting results.
For MKI67 we found no significant correlation (Supplementary Table S1) between Ki67 LI% and DNA methylation of CpGs linked with MKI67 promoter regions or within gene body. However, 3 consecutive island/promoter CpGs (cg26235537, cg10026577, cg02306970) were near significance after correction for the FDR and could constitute a VMR (Variably Methylated Region) of positive correlation (Spearman’s rho ~ 0.5). To our knowledge, a DMR (Differentially Methylated Region) or VMR with positive correlation within a regulatory region is linked to a tissue-specific mechanism and alternative transcription (Slieker).
For MCM6 we found 2 significant positive correlations within MCM6 gene body (cg10512742 with rho = 0.66 and cg15903064 with rho = 0.5) (Supplementary Table S2). These two loci could be linked to transcription robustness and stability mechanisms, and in fine to an increase of transcription. This is an interesting finding as these sites could constitute therapeutic targets in cancer (YANG).
Results (page 7, lines 235-243) and Discussion (page 8, lines 301-312) sections have been updated with these new data.
- The methylation class was not interpretable for quite large amount of samples (42%). Did you try to inspect lower filtering cut limit, e.g. 0.5? Also, how about global classifier MNP v11/12, were all tumor samples distinguished as meningioma?
When excluding only cases with a score lesser than 0.5 (4/33), 7 cases were classified as benign, 18 as intermediate, and 4 as malignant, without correlation with LI (MCM6 or Ki-67) and survival (added in Results, page 6, lines 232-235).
The global brain tumor classifier (MolecularNeuropathology.org) reported the methylation class “meningioma” with a valid score (calibrated score ˃ 0.9) for all samples (added in Results, page 6, lines 226-227).
- Figures 1&2 subparts lack suffixes in the text, e.g. Figure 1a,b,...
We added these suffixes in the text, in the results section.